

# Genome-wide identification of putative dihydroflavonol 4-reductase (*DFR*) gene family in eight Solanaceae species and expression analysis in *Solanum lycopersicum*

Wenjing Li[1,2,3,*], Yiming Zhang[1,*], Hualiang Liu[4], Qiuping Wang[1], Xue Feng[1], Congyan Wang[1], Yanxiang Sun[1], Xinye Zhang[1,2,3] and Shu Zhu[1]

[1] College of Life Science, Langfang Normal University, Langfang, Hebei, China
[2] Hebei Key Laboratory of Animal Diversity, Langfang, Hebei, China
[3] Langfang Key Laboratory of Cell Engineering and Application, Langfang, Hebei, China
[4] Xingtai University, Xingtai, Hebei, China
* These authors contributed equally to this work.

Corresponding author
Shu Zhu, zhushu@lfnu.edu.cn

## ABSTRACT

Dihydroflavonol 4-reductase (DFR; EC1.1.1.219) is an important rate-limiting enzyme in the plant flavonoid pathway toward both anthocyanins and proanthocyanidins. Although *DFR* genes have been isolated from multiple plants and their functions have been well characterized in some plants, little is known about *DFRs* in Solanaceae species. Therefore, in this study, we performed genome-wide analysis and identified 6, 5, 4, 5, 5, 6, 6 and 5 *DFR* gene family members in eight Solanaceae species (*S. lycopersicum*, *S. pennellii*, *S. tuberosum*, *S. melongena*, *C. annuum*, *N. tabacum*, *P. inflata*, and *P. axillaris*) respectively. The putative *DFR* genes were systematically identified using bioinformatics to predict their protein properties, cellular location, phylogenetic relationships, gene structure, conserved motifs, and *cis*-acting elements in the promoters. Furthermore, quantitative real-time PCR (qRT-PCR) was used to identify the expression pattern of *DFRs* in tomato. We classified all DFRs into five groups based on their phylogenetic features. Sequence analysis showed that all encoded DFR protein sequences possess a highly conserved NAD-dependent epimerase/dehydratase. In addition, almost all the members of each group displayed similar gene structures and motif distributions, which might be related to their identical executive functions. All 42 *DFRs* possess a series of light-responsive, phytohormone-responsive, MYB-responsive, stress-responsive, and tissue-specific expression-related *cis*-elements in the promoter sequences. qRT-PCR analysis showed that tomato *DFRs* were expressed in many different organs. This study will provide a theoretical basis for further investigation of the function of DFRs in Solanaceae.

## INTRODUCTION

In many plants, anthocyanins are major secondary metabolites that have been widely studied due to their important properties. They are involved in functions related to determining fruit and flower color and mitigating naturally occurring stresses to the plant (*Grotewold, 2006*; *Qiu et al., 2016*; *Kim et al., 2017*). Furthermore, anthocyanins were reported to delay over-ripening in tomato fruits and their over-expression resulted in a substantial increase in fruit shelf-life (*Bassolino et al., 2013*; *Zhang et al., 2013*). In recent years, the anthocyanin biosynthetic pathway has been well studied in several plants, such as *Arabidopsis thaliana* (*Gonzalez et al., 2008*), *Zea mays* (*Petroni & Tonelli, 2011*; *Pourcel et al., 2012*), *Petunia hybrida*, *Antirrhinum majus* (*Winkel-Shirley, 2001*), *Malus domestica* (*Espley et al., 2013*), and *Brassica oleracea* var. capitata f. rubra (*Sasaki, 2020*). Most of the genes involved in the flavonoid synthesis pathway have been identified, and it has been established that dihydroflavonol 4-reductase (DFR) is a pivotal multifunctional oxidoreductase involved in anthocyanin biosynthesis, which can selectively or unselectively catalyze the reduction of three colorless dihydroflavonols (DHFs)—dihydrokaempferol (DHK), dihydroquercetin (DHQ), and dihydromyricetin (DHM)—to their corresponding leucoanthocyanidins in an NADPH-dependent manner regulated by the MYB-bHLH-WD40 (MBW) complex (*Tian et al., 2017*). Leucoanthocyanidins are subsequently converted into their respective anthocyanidins and other flavonoids (*Li et al., 2017*).

To date, many *DFRs* have been cloned from multiple plants and *DFR* mutations have been shown to cause the loss of anthocyanins and proanthocyanidins in plants (*Zhu et al., 2018*). Although DFR proteins can catalyze the above three substrates in an appreciable number of plants, DHK cannot be used as a substrate in *Petunia* and *Cymbidium* species. As such, these species cannot produce pelargonidin-based orange flowers, indicating that DFRs from different species exhibit different substrate preferences (*Forkmann & Ruhnau, 1987*; *Johnson et al., 1999*). In view of its substrate specificity, DFR controls the flux into the three biosynthetic branches, leading to diverse anthocyanidins (cyanidin, delphinidin, pelargonidin) (*Forkmann & Ruhnau, 1987*; *Johnson et al., 1999*). The substrate preference of DFR from different plants can be determined by using recombinant proteins to analyze the enzyme activity (*Johnson et al., 2001*; *Katsu et al., 2017*; *Zhu et al., 2018*). Furthermore, we can elucidate the crystal structure to determine why most DFRs accept dihydroflavonols with different hydroxylation patterns. However, the purified DFR protein was described as very unstable; therefore, alignment of amino acid sequences—especially in the region responsible for substrate specificity—was the most effective way to determine the substrate preference (*Petit et al., 2007*).

Alignment and crystal structure studies characterized all DFRs containing an NADPH-binding Rossmann domain at the N-terminus and substrate-binding specificity in the variable C-terminus (*Petit et al., 2007*). The amino acid region from 131–156 has been characterized as the substrate-binding site. In particular, an asparagine (Asn, N) or aspartic acid (Asp, D) residue at position 134 has been shown to be associated with substrate recognition, although the variant N134D may not be specific only for recognizing

the three hydroxylation patterns in the B-ring of dihydroflavonols. Once we have identified the region that determines the substrate specificity of DFR, we can modulate the substrate specificity by mutating the amino acids in that region. To achieve this, scientists have generated chimeric *DFRs* using petunia *DFR*, which cannot reduce DHK, and gerbera *DFR*, which can reduce DHK, and introduced the chimeric *DFRs* to a mutant petunia line (*Johnson et al., 2001*). The first successful petunia flower color engineering was achieved using *DFR* cDNA cloned from maize (*Meyer et al., 1987*); Since then, multiple homologs have been cloned to modify anthocyanins (*Rosati et al., 2003*; *Davies et al., 2003*) and proanthocyanidins (*Bavage et al., 1997*; *Robbins et al., 1998*). The overexpression of functionally active DFR enzymes definitively increases anthocyanin accumulation in rice (*Takahashi et al., 2006*), tobacco (*Xie et al., 2004*), forsythia (*Rosati et al., 1997*), crabapple (*Tian et al., 2017*), and others.

To date, a few of gene families of Solanaceae, including argonautes (*Liao et al., 2020*), *WOX* (*Li et al., 2018*), and *SAUR* (*Wu et al., 2012*), have been identified by bioinformatics methods; in addition, the functions of some genes have been identified. However, despite many efforts, studies of the identification and expression pattern of *DFR* gene families in Solanaceae are scarce. In Solanaceae, different species display diverse fruit or flower colors due to various degrees of anthocyanin accumulation. In spiny solanums, variation in the DFR promoter region and the alternative splicing of DFR account for altered anthocyanin accumulation (*Wang et al., 2022*). In this study, we performed a systematic study to identify and characterize the *DFR* gene family in the genomes of eight Solanaceae species, including tomato (*Solanum lycopersicum*), wild tomato (*Solanum pennellii*), potato (*Solanum tuberosum*), eggplant (*Solanum melongena*), pepper (*Capsicum annuum*), tobacco (*Nicotiana tabacum*), and two petunia species (*Petunia inflata* and *Petunia axillaris*). Based on whole-genome sequencing results, the members of the putative *DFR* gene family in Solanaceae were identified by using bioinformatics analysis; subsequently, their sequence features, gene structure, evolutionary relationships, conserved motifs, chromosome distribution, cellular location, and *cis*-acting elements in the promoter were analyzed. Finally, the expression pattern of *DFR* in *S. lycopersicum* was analyzed by qRT-PCR methods. This fundamental research can provide a foundation for further research into the physiological and functional studies of the *DFR* family in *S. lycopersicum* and other related Solanaceae species.

## MATERIALS AND METHODS

### Plant materials and tissue collection

*Solanum lycopersicum* ('Micro-Tom') seeds were collected from Guangdong Ocean University and were sprouted in a greenhouse at 25 °C under a 16-h light/8-h dark cycle at Langfang Normal University. The seeds were sterilized for 10 min with 10% sodium hypochlorite and washing five times with sterile water. Five tomato major tissues—the 45-day-old seedling roots, stems, leaves, flowers, and green ripening fruits—were collected and conserved at −80 °C after liquid nitrogen treatment; three biological replicates were performed for each sample.

### *DFR* sequence retrieval and data analysis

The *DFR* sequences of *Arabidopsis thaliana*, *Solanum lycopersicum*, and *Vitis vinifera* were retrieved from Phytozome v13 (https://phytozome-next.jgi.doe.gov/) using the KEGG codes (K13082). The sequences were At5g42800.1, Solyc02g085020.4.1, and VIT_218s0001g12820.1 (*Kim et al., 2017*; *Zhu et al., 2018*; *Li et al., 2019*), which were previously reported and were used as queries to extract the *DFR* genes of eight Solanaceae species. The local BLAST program was performed against the genomic sequence of the eight Solanaceae species in the Solanaceae Genomics Network (www.solgenomics.net) with −5 expect (E) threshold. A total of 42 candidate members were found as listed in Table 1. All the candidate protein sequences were further checked for the presence of epimerase domains (PF01370) using the Pfam tool; the bit score between each member and at least two probes was not to be less than 240, and the candidate *DFR* genes were aligned to ensure that no gene was represented repeatedly. The number of amino acids, molecular weight (MW), isoelectric point (pI), instability index, and grand average of hydropathicity (GRAVY) index of the candidate DFR proteins were identified using ExPASy (http://web.expasy.org/protparam/). The cellular location was identified using CELLO v.2.5 (cello.life.nctu.edu.tw/).

### Phylogenetic analyses of the *DFR* gene family

The full-length DFR amino acid sequences of the eight species were aligned using MUSCLE or ClustalW (an inbuilt feature of MEGA 11.0) (*Edgar, 2004*). For phylogenetic analysis, the neighbor-joining (NJ) phylogenetic tree of the *DFR* gene family was constructed using MEGA 11.0 by performing 1,000 bootstraps (*Tamura, Stecher & Kumar, 2021*).

### Exon–intron structure and conserved motifs analysis

The exon-intron organization of the *DFR* genes were analyzed by comparing their respective coding and genomic sequence information in the Solanaceae Genomics Network database. Gene structure was presented using the Gene Structure Display Server (GSDS 2.0) (http://gsds.cbi.pku.edu.cn/) (*Hu et al., 2015*). Besides, MEME program 5.1.1 was used to identify finer motifs in the candidate DFR protein sequences (*Bailey et al., 2006*). The parameters were set as: site distribution, 0 or 1 site per sequence; number of motifs to find, 6; and width of the motif, 6–300 residues.

### *In silico* analysis of promoter sequences

To investigate the putative role of *cis*-acting elements that were responsible for gene expression, the upstream sequence (2,000 bp) of each coding sequence was retrieved from the Solanaceae Genomics Network. The sequences were analyzed by different bioinformatics programs, including PlantCARE (*Lescot et al., 2002*) and PLACE (*Higo et al., 1999*).

**Table 1** *DFR genes identified from eight sequenced Solanaceae genomes.*

| Index | Abbreviation | Species | Number of DFR genes |
|---|---|---|---|
| 1 | Sl | *Solanum lycopersicum* | 6 |
| 2 | Sp | *Solanum pennellii* | 5 |
| 3 | St | *Solanum tuberosum* | 4 |
| 4 | Sm | *Solanum melongena* | 5 |
| 5 | Ca | *Capsicum annuum* | 5 |
| 6 | Na | *Nicotiana attenuata* | 6 |
| 7 | Pi | *Petunia inflata* | 6 |
| 8 | Pa | *Petunia axillaris* | 5 |
| Total | | | 42 |

## Gene expression analysis of *SlDFRs* in tomato

To verify the expression profiles of six *SlDFR* genes, qRT-PCR was used to measure the expression of *SlDFRs* in different tomato tissue samples, including the roots, stems, leaves, flowers, and fruits. Total RNA was isolated using RNAsimple total RNA kit according to the manufacturer's recommendations (TIANGEN, Beijing, China). The quantity and quality of total RNA samples were tested using a Nanodrop spectrophotometer (Thermo, Waltham, MA, USA) and RNA gel electrophoresis. The DNase I-treated RNA was reverse-transcribed using GoScript™ Reverse Transcription System I (Promega, Madison, WI, USA), and qRT-PCR was performed using a CFX96 Touch thermocycler (Bio-Rad, Hercules, CA, USA). Gene-specific primers were designed using Primer 5.0 software to amplify 121–165 bp PCR products specific for each *SlDFR* gene (Table 2). The expression of the tomato *Actin* gene (GenBank accession no. FJ532351.1) was used as the internal control. Each reaction mixture contained 10 μL of 2 × SYBR Green qPCR Mix (Low ROX) (Aidlab, Beijing, China), 1.0 μL of diluted cDNA sample, and 400 nM of gene-specific primers in a final volume of 20 μL. The thermal cycling protocol used was as follows: 95 °C for 10 min, followed by 40 cycles of 95 °C for 15 s, 56 °C for 15 s, and 72 °C for 20 s. After the qRT-PCR reaction was completed, a melting curve was generated to analyze the specificity of each gene by increasing the temperature from 60 °C to 95 °C. Three independent biological replicates of the experiment were performed, and the significance was determined with IBM SPSS Statistics 20 software ($p \leq 0.05$).

## RESULTS

### Identification and physicochemical properties of *DFR* gene family members in Solanaceae species

We used the amino acid sequences of *DFR* genes in *Arabidopsis*, *Vitis bellula*, and *S. lycopersicum* as a target to query for related Solanaceae DFRs. Through comprehensive screening, 42 DFR protein sequences were retrieved, including six from tomato (*S. lycopersicum*), tobacco (*N. tabacum*), and petunia (*P. inflata*), four from potato (*S. tuberosum*), and five from wild tomato (*S. pennellii*), eggplant (*S. melongena*), pepper

**Table 2 The primers used in this study.**

| S. No | Name | Sequence | Product size (bp) |
|---|---|---|---|
| 1 | SlDFR | TTGGCTTGTCATGAGACTCC CCTTCCACTGCCAAGTCAGC | 150 |
| 2 | SlDFR like1-1 | GATCATGGCTCATCATGAGG CTCTCTGATGCACCTTCTAG | 121 |
| 3 | SlDFR like1-2 | CAGGATACCTAGCATCATGG TAGCCTTTGGGATGCTTCAG | 137 |
| 4 | SlDFR like2 | GGACAGATGTTGAGTTCTTG AAGATCAACACTACTTGGAG | 165 |
| 5 | SlDFR like3 | CAATGAGAGGTTGCATTGGC TTCAGAACATTCAAGGTCCC | 136 |
| 6 | SlDFR like4 | AAGGGTAAAGTATGTGTGAC GCTCCTTGTAGCTTCCATAG | 149 |

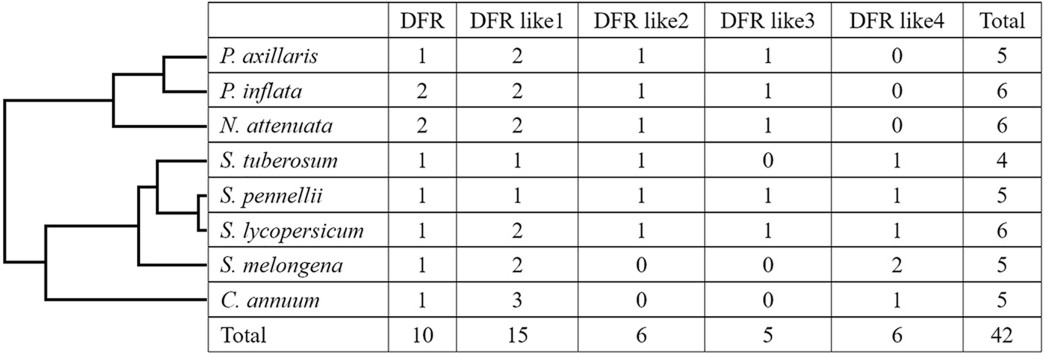

| | DFR | DFR like1 | DFR like2 | DFR like3 | DFR like4 | Total |
|---|---|---|---|---|---|---|
| P. axillaris | 1 | 2 | 1 | 1 | 0 | 5 |
| P. inflata | 2 | 2 | 1 | 1 | 0 | 6 |
| N. attenuata | 2 | 2 | 1 | 1 | 0 | 6 |
| S. tuberosum | 1 | 1 | 1 | 0 | 1 | 4 |
| S. pennellii | 1 | 1 | 1 | 1 | 1 | 5 |
| S. lycopersicum | 1 | 2 | 1 | 1 | 1 | 6 |
| S. melongena | 1 | 2 | 0 | 0 | 2 | 5 |
| C. annuum | 1 | 3 | 0 | 0 | 1 | 5 |
| Total | 10 | 15 | 6 | 5 | 6 | 42 |

**Figure 1 Distribution of the *DFR* genes in different plant species and groups.** The numbers in each column represent the number of genes in that species.

(*C. annuum*), and petunia (*P. axillaris*) (Table 1, Fig. 1). For convenience, all genes were designated as *DFR*, *DFR like1*, *DFR like2*, *DFR like3*, and *DFR like4* in different species, according to their similarity (Table 3).

Table 3 lists the 42 *DFR* genes and the proteins they encode, including gene name, length, molecular weight, isoelectric point, GRAVY index, instability index, cellular location, and chromosome start and end location (Fig. 1, Table 3). The number of amino acid of *DFR* genes ranged from 282 to 427 aa with an average of 340 aa. Molecular weights varied from 31,855.70 to 46,908.53 Da, and the isoelectric points were distributed from 5.23 to 9.18, which indicates that DFRs, except for PiDFR1, PiDFR2, PiDFR like3, and PaDFR like3, are all acidic proteins. A large divergence in GRAVY indices was observed, from −0.325 to 0.114, with the average GRAVY index of −0.18, −0.18, 0.08, −0.06, and −0.15 in five subfamilies, respectively. The instability index of these genes varied from 23.33 to 44.59 with an average of 34.63; most of the proteins were classified as stable proteins (83.3%). Most of DFRs located in cytoplasmic, except for PaDFR like2, SmDFR like4-1 and PiDFR2 located in periplasmic, and PaDFR like1-2 located in outer membrane.

**Table 3 The details of *DFR* gene family identified in Solanaceae.**

| S. No | Gene name | Protein ID | Protein/AA | Molecular weight/D | pI | GRAVY | Instability index | Cellular localization | Instability index | Location Chromosome/Scaffold | Start | End |
|---|---|---|---|---|---|---|---|---|---|---|---|---|
| 1 | *SlDFR* | Solyc02g085020.4.1 | 379 | 42,318.49 | 5.97 | −0.208 | 33.78 | Cytoplasmic | 33.78 | 2 | 46065285 | 46067689 |
| 2 | *SlDFR like1-1* | Solyc01g094070.3.1 | 324 | 36,655.29 | 6.10 | −0.029 | 30.5 | Cytoplasmic | 30.5 | 1 | 77915544 | 77917312 |
| 3 | *SlDFR like1-2* | Solyc05g051010.4.1 | 327 | 35,876.9 | 5.83 | −0.192 | 35.51 | Cytoplasmic | 35.51 | 5 | 60605195 | 60608354 |
| 4 | *SlDFR like2* | Solyc03g031470.3.1 | 334 | 36,717.69 | 6.67 | 0.114 | 23.33 | Cytoplasmic | 23.33 | 3 | 3917371 | 3919983 |
| 5 | *SlDFR like3* | Solyc12g005350.2.1 | 341 | 37,945.82 | 6.46 | −0.040 | 39.67 | Cytoplasmic | 39.67 | 12 | 229271 | 231650 |
| 6 | *SlDFR like4* | Solyc04g008780.4.1 | 338 | 37,872.52 | 6.2 | −0.169 | 34.96 | Cytoplasmic | 34.96 | 4 | 2449106 | 2452152 |
| 7 | *SpDFR* | Sopen02g029720.1 | 382 | 42,668.96 | 6.2 | −0.238 | 31.82 | Cytoplasmic | 31.82 | 2 | 52012315 | 52014356 |
| 8 | *SpDFR like1* | Sopen01g037880.1 | 329 | 37,129.86 | 5.95 | −0.007 | 30.71 | Cytoplasmic | 30.71 | 1 | 95912119 | 95913769 |
| 9 | *SpDFR like2* | Sopen03g005360.1 | 334 | 36,635.52 | 6.4 | 0.088 | 24.99 | Cytoplasmic | 24.99 | 3 | 4184663 | 4187477 |
| 10 | *SpDFR like3* | Sopen12g001330.1 | 341 | 38,022.91 | 6.86 | −0.064 | 38.27 | Cytoplasmic | 38.27 | 12 | 268582 | 271680 |
| 11 | *SpDFR like4* | Sopen04g003960.1 | 338 | 37,871.72 | 6.67 | −0.116 | 32.94 | Cytoplasmic | 32.94 | 4 | 2634215 | 2637585 |
| 12 | *StDFR* | PGSC0003DMT400009287 | 382 | 42,469.78 | 5.71 | −0.171 | 31.77 | Cytoplasmic | 31.77 | 2 | 40293862 | 40297510 |
| 13 | *StDFR like1* | PGSC0003DMT400018059 | 331 | 37,039.54 | 5.84 | −0.179 | 35.59 | Cytoplasmic | 35.59 | 5 | 45576555 | 45580152 |
| 14 | *StDFR like2* | PGSC0003DMT400013000 | 338 | 36,959.81 | 5.93 | 0.087 | 23.48 | Cytoplasmic | 23.48 | 3 | 2354247 | 2357891 |
| 15 | *StDFR like4* | PGSC0003DMT400065107 | 335 | 37,456.07 | 6.67 | −0.147 | 36.52 | Cytoplasmic | 36.52 | 4 | 3924805 | 3927853 |
| 16 | *PiDFR1* | Peinf101Scf00073g04027.1 | 353 | 39,450.55 | 7.13 | −0.021 | 40.58 | Cytoplasmic | 40.58 | Peinf101Scf00073 | 448400 | 450165 |
| 17 | *PiDFR2* | Peinf101Scf00590g21022.1 | 282 | 31,855.70 | 8.43 | −0.163 | 30.70 | Periplasmic | 30.70 | Peinf101Scf00590 | 2119809 | 2135294 |
| 18 | *PiDFR like1-1* | Peinf101Scf01326g04016.1 | 297 | 33,015.64 | 6.00 | −0.205 | 36.04 | Cytoplasmic | 36.04 | Peinf101Scf01326 | 438710 | 440897 |
| 19 | *PiDFR like1-2* | Peinf101Scf01326g03006.1 | 298 | 33,006.56 | 6.25 | −0.253 | 34.24 | Cytoplasmic | 34.24 | Peinf101Scf01326 | 321752 | 323931 |
| 20 | *PiDFR like2* | Peinf101Scf00140g08006.1 | 313 | 33,980.93 | 5.31 | 0.068 | 28.89 | Cytoplasmic | 28.89 | Peinf101Scf00140 | 817602 | 819839 |
| 21 | *PiDFR like3* | Peinf101Scf01299g00011.1 | 354 | 39,518.82 | 9.18 | −0.104 | 44.24 | Cytoplasmic | 44.24 | Peinf101Scf01299 | 73147 | 79583 |
| 22 | *PaDFR* | Peaxi162Scf00366g00630.1 | 353 | 39,389.42 | 6.37 | −0.033 | 41.98 | Cytoplasmic | 41.98 | Peaxi162Scf00366 | 655201 | 656979 |
| 23 | *PaDFR like1-1* | Peaxi162Scf00238g00125.1 | 316 | 35,226.19 | 6.82 | −0.325 | 36.17 | Cytoplasmic | 36.17 | Peaxi162Scf00238 | 1258582 | 1260813 |
| 24 | *PaDFR like1-2* | Peaxi162Scf00238g01311.1 | 427 | 46,908.53 | 6.43 | −0.201 | 38.78 | OuterMembrane | 38.78 | Peaxi162Scf00238 | 1315303 | 1320945 |
| 25 | *PaDFR like2* | Peaxi162Scf00781g00211.1 | 313 | 33,826.80 | 5.54 | 0.094 | 28.60 | Periplasmic | 28.60 | Peaxi162Scf00781 | 212259 | 214581 |

(Continued)

| S. No | Gene name | Protein ID | Protein/AA | Molecular weight/D | pI | GRAVY | Instability index | Cellular localization | Instability index | Location Chromosome/Scaffold | Start | End |
|---|---|---|---|---|---|---|---|---|---|---|---|---|
| 26 | *PaDFR like3* | Peaxi162Scf00000g00839.1 | 355 | 39,421.49 | 8.58 | −0.107 | 43.60 | Cytoplasmic | 43.60 | Peaxi162Scf00000 | 8346643 | 8351171 |
| 27 | *CaDFR* | PHT91252 | 348 | 39,223.98 | 5.91 | −0.3 | 31.38 | Cytoplasmic | 31.38 | 2 | 155915163 | 155916746 |
| 28 | *CaDFR like1-1* | PHT82354 | 327 | 36,381.85 | 6.6 | −0.135 | 32.87 | Cytoplasmic | 32.87 | 5 | 233082910 | 233084595 |
| 29 | *CaDFR like1-2* | PHT65511 | 327 | 36,758.21 | 6.27 | −0.181 | 40.15 | Cytoplasmic | 40.15 | 12 | 42280995 | 42287068 |
| 30 | *CaDFR like1-3* | PHT93586 | 331 | 37,098.48 | 6.42 | −0.162 | 32.03 | Cytoplasmic | 32.03 | 1 | 60090121 | 60092229 |
| 31 | *CaDFR like4* | PHT81181 | 340 | 37,969.55 | 6.03 | −0.189 | 39.46 | Cytoplasmic | 39.46 | 5 | 23953961 | 23959080 |
| 32 | *SmDFR* | SMEL_000g030720.1.01 | 382 | 42,572.61 | 5.46 | −0.231 | 34.83 | Cytoplasmic | 34.83 | SMEL3CH00.06499 | 2363470 | 2365434 |
| 33 | *SmDFR like1-1* | SMEL_005g225130.1.01 | 332 | 36,652.86 | 5.94 | −0.190 | 31.98 | Cytoplasmic | 31.98 | SMEL3CH05 | 1716918 | 1720087 |
| 34 | *SmDFR like1-2* | SMEL_004g205360.1.01 | 305 | 34,380.93 | 5.23 | −0.242 | 39.74 | Cytoplasmic | 39.74 | SMEL3CH04 | 16495633 | 16500722 |
| 35 | *SmDFR like4-1* | SMEL_001g150260.1.01 | 328 | 35,877.25 | 6.96 | −0.069 | 30.55 | Periplasmic | 30.55 | SMEL3CH01 | 131345410 | 131350921 |
| 36 | *SmDFR like4-2* | SMEL_011g375900.1.01 | 351 | 39,425.29 | 6.76 | −0.196 | 37.52 | Cytoplasmic | 37.52 | SMEL3Ch11 | 66373647 | 66376655 |
| 37 | *NaDFR1* | OIS98434 | 349 | 39,069.87 | 6.42 | −0.197 | 42.96 | Cytoplasmic | 42.96 | 9 | 9300451 | 9307387 |
| 38 | *NaDFR2* | OIT31825 | 380 | 42,282.40 | 5.87 | −0.202 | 32.38 | Cytoplasmic | 32.38 | scaffold01190 | 169105 | 171023 |
| 39 | *NaDFR like1-1* | OIT07851 | 331 | 36,723.15 | 6.32 | −0.138 | 32.55 | Cytoplasmic | 32.55 | 1 | 37268172 | 37270851 |
| 40 | *NaDFR like1-2* | OIT29772 | 328 | 36,625.86 | 6.27 | −0.214 | 35.57 | Cytoplasmic | 35.57 | scaffold01693 | 184391 | 186705 |
| 41 | *NaDFR like2* | OIT35484 | 329 | 35,865.36 | 6.25 | 0.032 | 28.11 | Cytoplasmic | 28.11 | scaffold00554 | 418106 | 420780 |
| 42 | *NaDFR like3* | OIT01988 | 365 | 40,015.94 | 6.01 | −0.033 | 44.59 | Cytoplasmic | 44.59 | 6 | 33244851 | 33250819 |
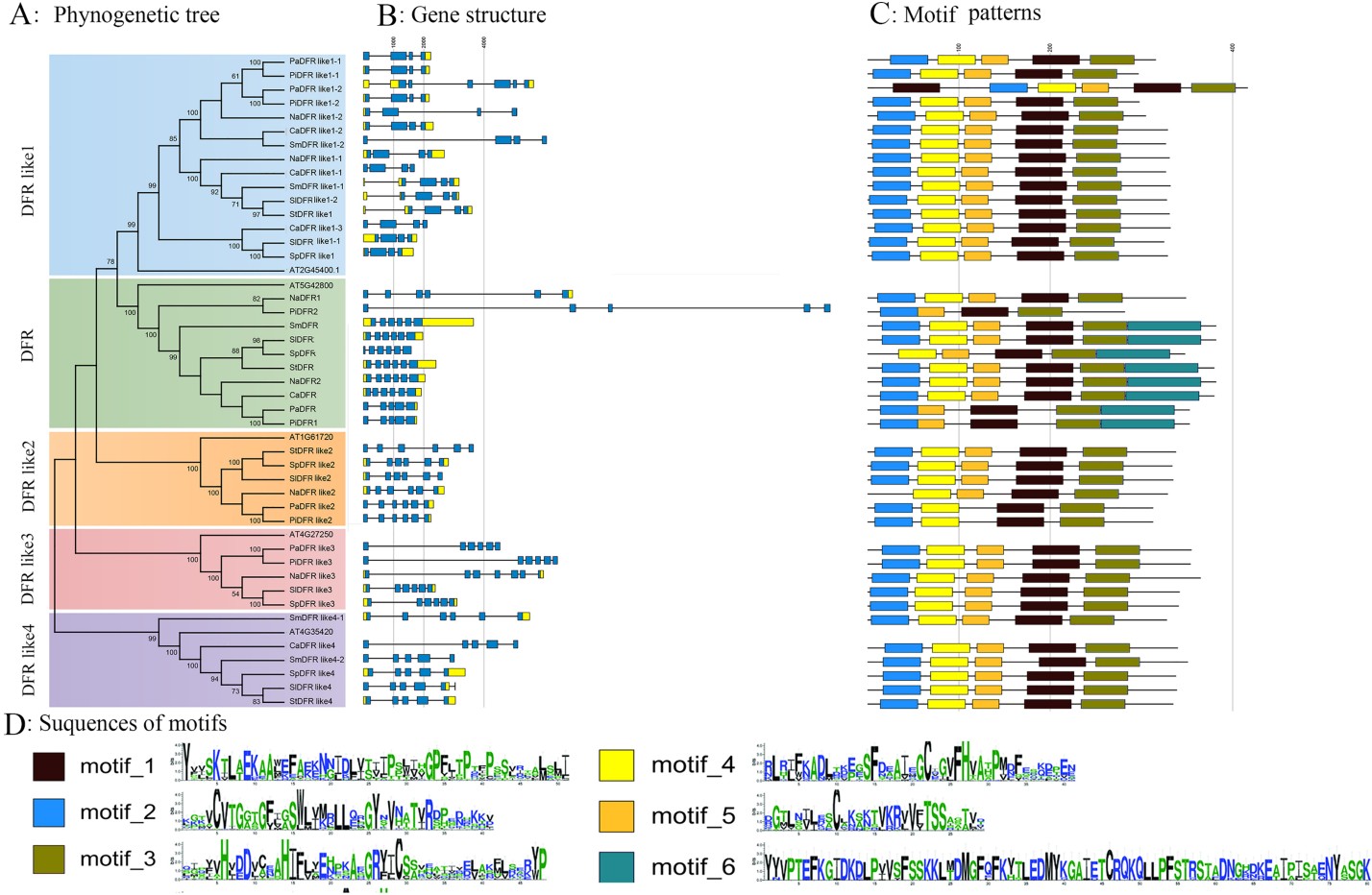

**Figure 2 Phylogenetic relationship, gene structure, and composition of conserved motifs of the *DFR* genes in Solanaceae plants.** (A) The phylogenetic tree was constructed using neighbour-joining method with 1,000 bootstrap replicates by MEGA11.0, and the bootstrap values >50 were indicated. The five major groups (DFR, DFR like1-4) were marked with different coloured backgrounds. Sl, Sp, St, Sm, Pi, Pa, Ca, Na in (A) representing *S. lycopersicum*, *S. pennellii*, *S. tuberosum*, *S. melongena*, *P. inflata*, *P. axillaris*, *C. annuum*, and *N. tabacum* respectively. (B) Exon/intron structures of *DFRs* from Solanaceae plants. The UTR, exons were marked with yellow and blue boxes, respectively. The introns were marked with black lines. (C) Motif composition of the *DFR* proteins in plants. Each motif was indicated by a coloured boxes numbered at the bottom. (D) Sequence logos of the six motifs identified using the MEME search tool (E-value < 0.00001). The height of the letter represents the degree of amino acid conservation at the corresponding position. The numbers on the x-axis and y-axis represent the residue positions in the motifs and the information content measured in bits, respectively.

Although the DNA length of *DFRs* varied over a wide range, the length of each CDS and protein was similar within each subfamily.

### Gene structure and phylogenetic analysis of the *DFR* gene family in Solanaceae

To investigate the phylogenetic relationships of DFR proteins, a NJ tree was constructed based on the full length of all 42 DFR sequences from the eight species listed in Table 1. As shown in the phylogenetic tree (Fig. 2A), all 42 proteins were clustered into five groups (DFR, DFR like1, DFR like2, DFR like3, and DFR like4) with high bootstrap values consisting of 10, 15, six, five and six members, respectively. It was found that *S. lycopersicum* and *S. pennellii* contained all five groups of DFR proteins, *S. tuberosum,*

*Nicotiana attenuata*, *P. inflata*, and *P. axillaris* contained four, and *S. melongena* and *C. annuum* contained three. With the exception of *N. attenuata* and *P. inflata*, which contained two DFR proteins, all species contained one DFR protein. The results above suggested that the Solanaceae *DFRs* were derived from one ancestor gene and that they developed into different branches after their lineages diverged.

In addition, to gain an insight into the variation in the *DFR* genes, we analyzed the exon–intron structure. The structure of the *DFR* genes was relatively conserved within each subfamily (Fig. 2B). The number of exons ranged from 4 to 6; among them, six, five, and four exons were identified in 20, 8, and 14 genes, respectively. The *DFR like1* subfamily was characterized by four exons (93.3%), except for *PaDFR like1-2*, which had six exons. The *DFR* subfamily was characterized by six exons (70%), similar to other *DFR* genes from arabidopsis, petunia, snapdragon, morning glory, and onion plants, except for *PiDFR2*, *PaDFR*, *PiDFR1*, which had five exons. The *DFR like4* subfamily featured five exons (83.3%), except for *CaDFR like4*. Both the *DFR like2* and *DFR like3* subfamilies featured five exons (100%) (Fig. 2B). The number of exons within each subfamily agreed with that of *DFRs* in the tea plant (*Mei et al., 2019*). The conservation of the *DFR* gene structure revealed the ancient features of the evolution.

## Analysis of conserved motifs in DFR proteins

The patterns of conserved motifs were predicted using MEME5.1.1, six conserved motifs in Solanaceae were captured (Fig. 2C). Motif 1 (175–225 aa in SlDFR labeling) amounted to the NAD-dependent epimerase/dehydratase family, Motif 2 (19–60 aa) encoded a NAD(P)H-binding domain, Motif 5 (117–145 aa) corresponded to 3-beta hydroxysteroid dehydrogenase/isomerase family, Motif 6 (282–362 aa) encoded a domain of unknown function (DUF1731), and Motif 3 (234–281 aa) and 4 (69–109 aa) did not match any functional annotation (Fig. 2D). Of the 42 DFRs, all the proteins contained Motif 1 and 3; CaDFR and NaDFR like2 had lost Motif 2; PaDFR, PiDFR1, and PiDFR2 had lost Motif 4; and PaDFR like1-1, PiDFR like1-1, NaDFR like1-2, CaDFR like1-2, SlDFR like1-1, PaDFR like2, and PiDFR like2 had lost Motif 5. Only the DFR subfamily contained Motif 6, except for NaDFR1 and PiDFR2. SlDFR contained the above six conserved motifs, similar to VvDFR.

Multiple sequence alignment of Solanaceae DFR proteins was carried out using Genedoc software. All of the six SlDFR proteins contain conserved NADPH-binding domains, showing that they belong to the NAD-dependent epimerase/dehydratase family. Consistant with DFRs in other plants, only SlDFR possessed a conserved substrate specificity-determining region (Fig. 3), which showed that maybe SlDFRs was unique. The 138[th] asparagine residue (ZmDFR lableing, *i.e.*, N133 of VvDFR in Fig. 3) is said to be extremely important for choosing substrate. However, in Solanaceae plants, N is substituted for D. Thus, SlDFR fell into Asp-type DFR, which converts DHK inefficiently. The remaining putative SlDFRs are neither Asn nor Asp types.

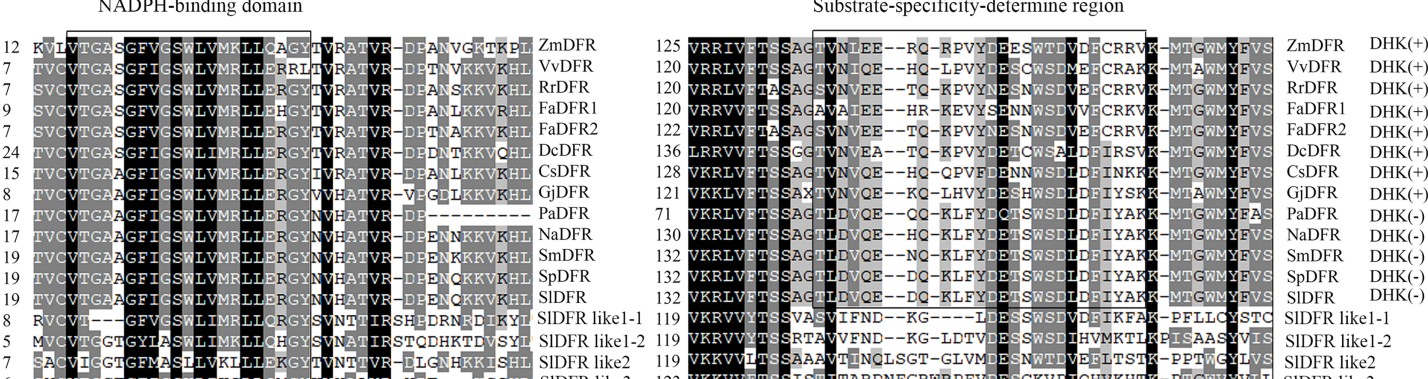

**Figure 3 Multiple alignment analysis of amino acid sequence of putative Solanaceae DFRs (utilizing DHK as substrate or not).** The numbers on the left represent residuals of different DFRs. DHK (+) or DHK (−) indicate whether these typical DFRs accept DHK as substrate (Asp-type) or not (Asn-type). The shading in different colors indicates the conserved percent of amino acid residues. The accession numbers of the protein sequences are as follows: Zm (*Zea mays*), NP_001152467.2; Vv (*Vitis vinifera*), CAA53578.1 or P93799; Rr (*Rosa rugosa*), ALR74719.1; Fa (*Fragaria X ananassa*), AHL46444.1 (FaDFR1), AHL46451.1 (FaDFR1); Dc (*Dianthus caryophyllus*), P51104.1; Cs (*Camellia sinensis*), AB018685.1; Gj (*Gerbera jamesonii*), AHF58605.1; Pa (*Petunia axillaris*), Peaxi162Scf00366g00630.1; Na (*Nicotiana attenuata*), OIT31825; Sm (*Solanum melongena*), SMEL_000g030720.1.01; Sp (*Solanum pennellii*), Sopen02g029720.1; Sl (*Solanum lycopersicum*), Solyc02g085020.4.1 (SlDFR), Solyc01g094070.3.1 (SlDFR like1-1), Solyc05g051010.4.1 (SlDFR like1-2), Solyc03g031470.3.1 (SlDFR like2), Solyc12g005350.2.1 (SlDFR like3), Solyc04g008780.4.1 (SlDFR like4).

## Tissue specificity of tomato *DFRs*

In order to get some idea of where the *DFR* genes function in the plant, the expression of six *SlDFR* genes was examined by means of qRT-PCR in five different organs: 45-day-old seedling root, stem, leaf, flower, and green ripening fruit. The expression profiles of each gene greatly differed (Fig. 4). Results showed that the six *SlDFR* genes were widely expressed in different organs at both the seedling stage and reproductive growth stage. This expression pattern reflected their physiological functions in each tissue. Among these genes, *SlDFR*, *SlDFR like1-1*, and *SlDFR like1-2*, with similar expression patterns, displayed high expression in stem, leaf, and flower; moreover, *SlDFR like1-2* showed high expression in root. *SlDFR like2* was preferentially expressed in the flowers. The expression of *SlDFR like3* was relatively high in all organs, except for the leaf. Meanwhile, transcripts of *SlDFR like4* were detected in all organs that we examined. These results suggested that there was a high correlation between qRT-PCR data and the data from the database.

## *Cis*-elements in the promoter sequences of *DFR* genes

To identify putative *cis*-elements in the promoter region of *DFRs*, genomic sequences located approximately 2,000 bp from the translational start site were analyzed in the PlantCARE and PLACE databases. The locations of *cis*-regulatory elements in the promoter sequences of *DFRs* were predicted to understand the possible roles of *DFRs* in response to abiotic stresses (Fig. 5). There were 48 *cis*-elements with 12 types in *PaDFR's* promoter region, which contained the most among all the putative *DFRs*. For other *DFRs*, 11–45 elements with 5–15 types were found (File S1). All these promoters, especially *PiDFR like1-1* and *StDFR*, were rich in light-responsive elements, such as GT-1 motif, Box II, Box4, AE box, I-Box, Sp1, ABRE, ABRE4, G-Box, ATC-motif, GA-motif, TCT-motif,

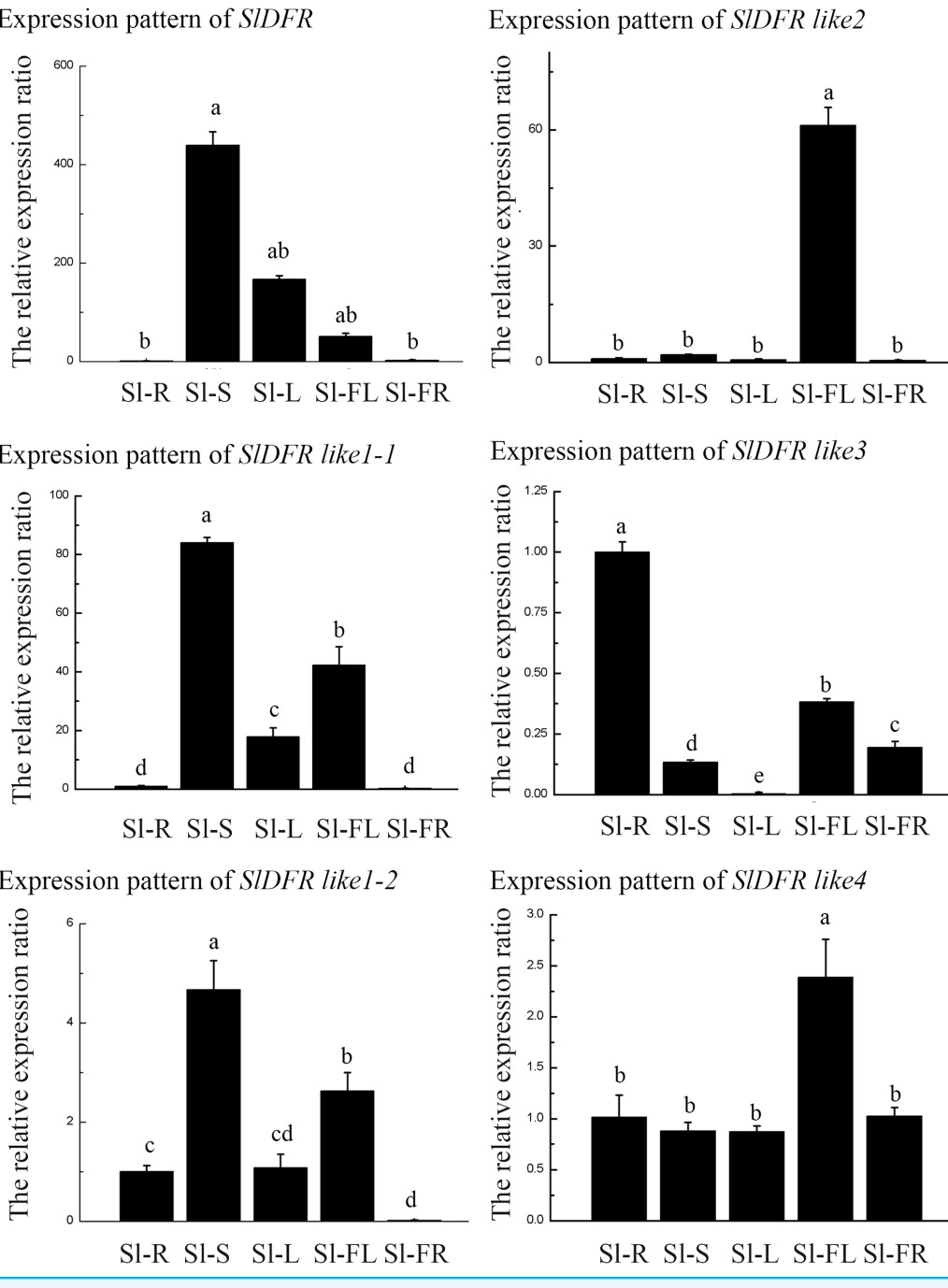

**Figure 4 The relative expression ratio of *SlDFR* genes.** The name of the gene was written on the top of each bar diagram (error bars indicate the standard deviation from three replicates). Sl-R, Sl-S, Sl-L, Sl-FL, Sl-FR in the X-axis representing tomato roots, stems, leaves, flowers, and fruits respectively. Different lowercase letters indicate significantly different values at *P* < 0.05 (least significant difference, LSD).

and GATA-motif. Furthermore, some promoters had several MYB and MYC elements. *StDFR*, *PaDFR like1-1*, and *PiDFR like1-1* contained two MYB-binding sites (CAACAG), and *PaDFR like3*, *PaDFR like1-2*, *PaDFR like2*, *PiDFR1*, *PiDFR2*, *SmDFR like1-1*, and *SpDFR like2* contained one MYB-binding site. *PaDFR* contained three MYB-recognition

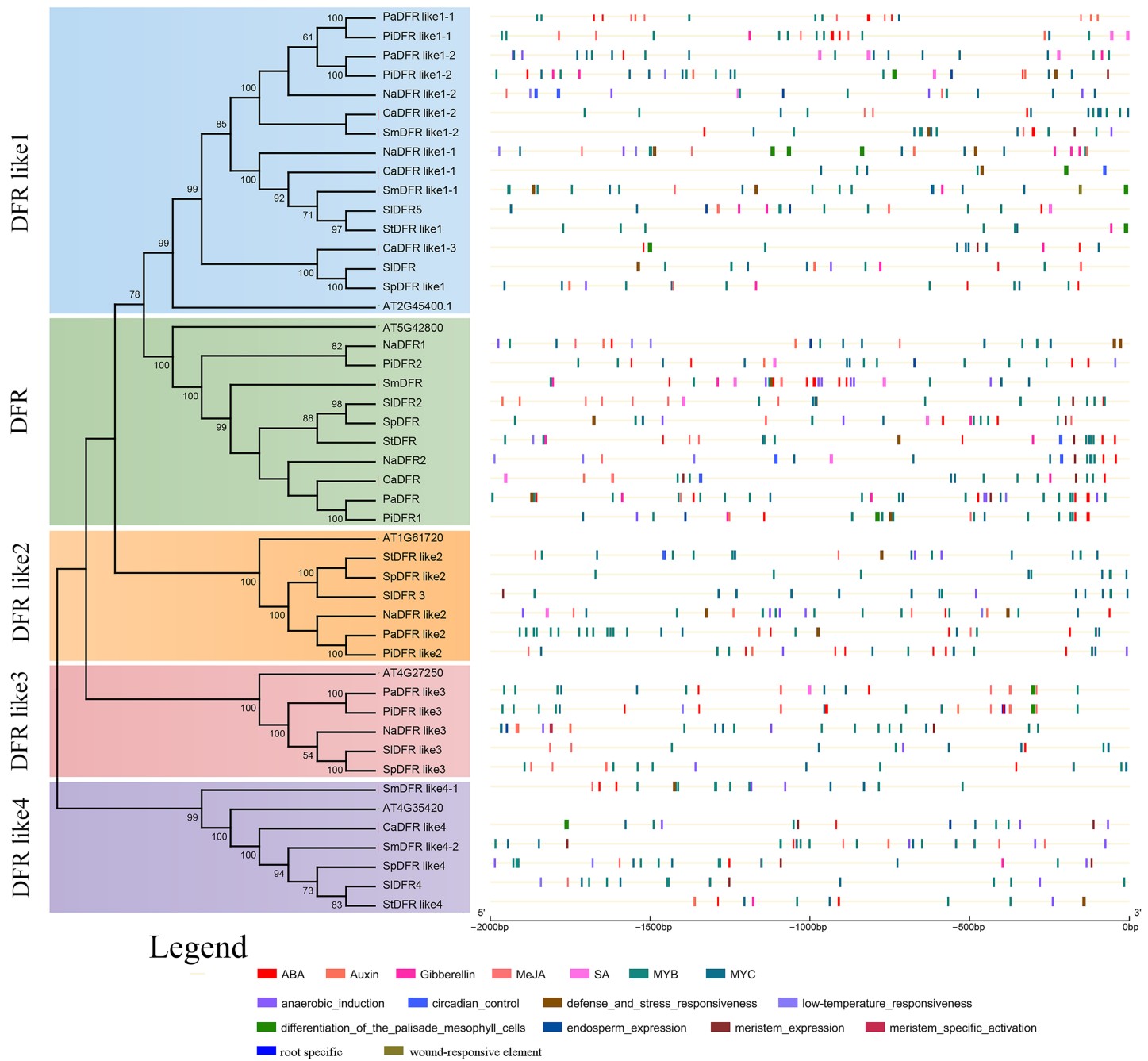

**Figure 5 Predicted *cis*-regulatory elements in the promoters of Solanaceae *DFR* genes.** The phylogenetic tree of the *DFR* family is replotted from Fig. 2. The *cis*-regulatory elements (CREs) in the 2 kb upstream regions of the 42 Solanaceae *DFR* genes were predicted using the PlantCARE and PLACE database. Black lines indicate promoter regions. CREs involved in response to phytohormones and induction of tissue specific expression are represented by color boxes.

sites (CCGTTG), *PiDFRlike1-1* contained two MYB-recognition sites, *NaDFR like2, SmDFR* and *PaDFR like3* contained one MYB-recognition site. In addition, *cis*-elements involved in root-specific expression were only found in the promoter of *PiDFR like3*.

Figure 5 presents the distribution and numbers of *cis*-elements responding to phytohormones, stresses, and tissue-specific expression.

## DISCUSSION

Recent studies have suggested that *DFR*s in plant species are encoded by a gene family, and studies of the *DFR* gene family focusing on *Brassica rapa* (*Ahmed et al., 2014*), *Freesia hybrida* (*Li et al., 2017*), tea plant (*Mei et al., 2019*) and *Brassica napus* (*Qian et al., 2023*) have been reported. Different DFR copies are responsible for diverse functions. For example, different copies can be expressed in different tissues or at different times, or different but related substrates could be used to form three products. However, there have been no reports about the family members and functions of DFRs in Solanaceae plants. This study systematically identified 42 putative DFRs within the eight Solanaceae species and performed genome-wide identification and phylogenetic analysis and determined the gene structure, conserved motifs, expression patterns, cellular location, and *cis*-acting elements. The Solanaceae DFR members, together with five *Arabidopsis* DFR proteins, were divided into five subfamilies (DFR, DFR like1, DFR like2, DFR like3, and DFR like4) based on the phylogenetic analysis (Fig. 2).

The analysis of the physicochemical properties of the protein and the number and MW of amino acids were found to be quite different from each other, indicating some differences in their structure and function. The number of introns ranged from three to five, which was consistant with the results that the firstly cloned maize *DFR* contained three introns, while *Petunia hybrida* and *Antirrhinum majus DFR* contained five introns (*O'Reilly et al., 1985*; *Beld et al., 1989*). Previous studies have shown that different DFR subtypes in the same plant share 25.5%–59.6% amino acid sequence identity (*Mei et al., 2019*). In this study, the alignment of the 42 protein sequences was performed by CLUSTAL-W using MEGA 11.0. The results revealed that the DFRs of Solanaceae had little homology. Within a single species (*e.g.*, *S. lycopersicum*), SlDFR shared 42.41%, 36.8%, 40.18%, 39.52%, and 37.67% identity with SlDFR like1-1, SlDFR like1-2, SlDFR like2, SlDFR like3, and SlDFR like4, respectively (Table S1). Although the sequences of tomato DFR like proteins greatly differ from SlDFR, they all have a conserved NADPH-binding domain, but only SlDFR has the substrate-binding domain (Fig. 3). Therefore, it is unclear whether they have the ability to form leucoanthocyanidins, similar to the outcomes of a previous study (*Mei et al., 2019*). Only one typical SlDFR has been reported so far, and the gene sequence was similar to that previously reported (*Bongue-Bartelsman et al., 1994*). Furthermore, a homology of over 96.04% among different plants was observed, such as tomato and potato, suggesting that these genes are highly conserved. Collectively, although 4–6 DFRs were found in every species of Solanaceae plants, only 1–2 DFRs were typical. For the other putative *DFR* proteins, further investigation is necessary.

DFR proteins can catalyze DHK, DHQ, and DHM to form their corresponding leucoanthocyanidins in many plants, such as *Gerbera* (*Johnson et al., 2001*), *Z. mays* (*Meyer et al., 1987*), *V. vinifera* (*Sparvoli et al., 1994*), and *V. bellula* (*Zhu et al., 2018*). However, *Petunia* and *Cymbidium* DFRs cannot reduce DHK efficiently, indicating that DFRs from different species exhibit diverse substrate preferences (*Gerats et al., 1982*;

*Forkmann & Ruhnau, 1987*; *Johnson et al., 1999*). Homologous sequence alignment is an effective method to determine the relationships between the substrate preference and amino acids in the region responsible for substrate specificity. It has been reported that in *Gerbera* DFR, residues 134 and 145 play important roles in the substrate specificity (*Johnson et al., 2001*). The glutamic acid (Glu, E) at position 145 is conserved in almost all DHK-accepting dicot DFRs, and its mutation (Glu to Leu) results in white flowers, although this is not the case for petunia DFR. In this article, all DFRs have E at residue 149, except for PaDFR. Recent investigations support that the presence of N at position 134 would determine the acceptance of DHK as substrate, whereas those with an D have a marked preference for DHQ (*Gosch et al., 2014*); in addition, a different mutation at site 134 changed the preference of DFR and modified its flux-controlling role. The *Lathyrus japonicus* and *Medicago truncatula* DFRs containing N at position 133, reduce DHK more efficiently than DHQ, while *Petunia* and *Cymbidium* DFRs containing a D at the same position, reduce DHK inefficiently. As the assumption mentioned above, whether the amino acid sequence at the specific location determining the substrate specificity should be verified in other ways. All the Solanaceae DFRs listed here have a D residue at position 145 (tomato numbering), suggesting that perhaps all the Solanaceae plants used DHQ and DHM as a substrate (Fig. 3).

Promoters of all the *DFR* genes of the eight Solanaceae species were screened for *cis*-regulatory elements, and 11–48 elements of 5–15 types were found. Among them, hormone-responsive, light-responsive, abiotic stress-responsive, development stage-related, and MYB-responsive elements were found, suggesting that *DFR* genes may play important roles in plant development and adaptation to environmental conditions. To date, different transcriptional studies have shown that the expression of DFR is regulated by different factors. A single SNP at −301 in *S. melongena DFR* promoter, which belongs to MYB recognition site (CCGTTG) at the second nucleotide, influenced the interaction of the *DFR* promoter with MYB113, affecting the transcription of *DFR* in *S. melongena*, thereby abolishing anthocyanin production in this species (*Wang et al., 2022*). It has reported that the mutations in *MYB113* rather than in *DFR* causing a decrease in expression of both *MYB113* and *DFR* in the *S. melongena* accessions with purple flowers and green or white fruit peels (*Babak et al., 2020*). Most domesticated tomato cultivars lack of anthocyanins in tomato fruits, contain a splicing site variation in *SlAN2-like* that causes the production of a non-functional SlAN2-like protein (*Colanero et al., 2020*; *Colanero, Perata & Gonzali, 2020*; *Sun et al., 2020*). Furthermore, all the tomato DFRs lacking MYB recognition site is the likely cause for the anthocyanin-free phenotype of tomato. High temperatures during night-time and a synthetic auxin, 2,4-dichlorophenoxyacetic acid, were found to inhibit the expression of *VvDFR* (*Mori, Sugaya & Gemma, 2005*; *Ban et al., 2003*). Light and abscisic acid were found to induce activation of the DFR promoter (*Ban et al., 2003*). The identification of the expression profiles will be useful for the classification of genes involved in the regulation of the precise nature of individual tissue. The expression profiles of *SlDFRs* revealed in this study agree with those in previous publications.

## CONCLUSIONS

In this study, we performed a comprehensive genome-wide analysis of *DFRs* in eight Solanaceae species. A total of 42 DFRs were identified, and they could be divided into five subfamilies. Analysis of the sequences and comparison of these genes with *DFR* genes from other species validated them as *DFR* genes. After analysis of phylogenetics and conserved motifs, we found that almost all of the DFR proteins contained NAD-dependent epimerase/dehydratase and an NAD(P)H-binding domain, but only the functional DFR proteins had substrate specificity-determining regions. The analysis of *cis*-acting elements revealed that the *DFRs* from eight Solanaceae species were involved in hormone response, light induction response, abiotic stress, and development stages. The expression patterns of the selected *SlDFR* genes were shown to be different in diverse tomato tissues. The elaborate results conferred here would provide valuable data for the useful research and application of DFR family in Solanaceae plants.

### Funding

This work was financially supported by the Fundamental Research Funds for the Universities in Hebei Province (Grant No. JYQ202101), the Science and Technology Project of Hebei Education Department (Grant No. ZD2020122), and the Langfang Normal University Undergraduate innovation and entrepreneurship training program fund (Grant No. S202010100016). The funders had no role in study design, data collection and analysis, decision to publish, or preparation of the manuscript.

### Grant Disclosures

The following grant information was disclosed by the authors:
Fundamental Research Funds for the Universities in Hebei Province: JYQ202101.
Science and Technology Project of Hebei Education Department: ZD2020122.
Langfang Normal University Undergraduate Innovation and Entrepreneurship Training program: S202010100016.

### Competing Interests

The authors declare that they have no competing interests.

### Author Contributions

- Wenjing Li conceived and designed the experiments, prepared figures and/or tables, and approved the final draft.
- Yiming Zhang performed the experiments, prepared figures and/or tables, and approved the final draft.
- Hualiang Liu performed the experiments, prepared figures and/or tables, and approved the final draft.
- Qiuping Wang performed the experiments, prepared figures and/or tables, and approved the final draft.

- Xue Feng analyzed the data, authored or reviewed drafts of the article, and approved the final draft.
- Congyan Wang analyzed the data, authored or reviewed drafts of the article, and approved the final draft.
- Yanxiang Sun analyzed the data, authored or reviewed drafts of the article, and approved the final draft.
- Xinye Zhang analyzed the data, authored or reviewed drafts of the article, and approved the final draft.
- Shu Zhu conceived and designed the experiments, prepared figures and/or tables, and approved the final draft.

## Data Availability
The raw measurements are available in the Supplemental Files.

## Supplemental Information
Supplemental information for this article can be found online at http://dx.doi.org/10.7717/peerj.16124#supplemental-information.

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
