# Peer review of "Genome-wide identification of putative dihydroflavonol 4-reductase (*DFR*) gene family in eight Solanaceae species and expression analysis in *Solanum lycopersicum"

_PeerJ, doi:10.7717/peerj.16124_

## Round 0.1 · original submission · Major Revisions

The detailed bioinformatics analysis of DFR gene family is acceptable without functional validation. The use of Phytozome or Gramene database is also acceptable as these resources are well-developed comparative genomic resources. So, authors can ignore the comments regarding these two issues. However, It may help to improve manuscript if authors can discuss how this study helped to improve the functional gene annotation of the DFR gene family and if they have any plan to submit those annotations to a public, open-source database (For example see a previously published paper in PeerJ: https://peerj.com/articles/11052). I look forward to revision of this manuscript with addressing all the comments made by three reviews.

Reviewer 1 ·

Basic reporting

In this study, authors have performed the genome wide identification and analysis of DFR gene family in several Solanaceae species. Moreover, they also performed the expression analysis of DFR genes of tomato. I believe that this work gives the fundamental knowledge about solanaceous plants' DFR genes. Nevertheless, the conclusions drawn from this study are overly simplistic, and much more investigation is required before this work can be accepted to PeerJ. Authors used Phytozome database for identifying the DFR genes in different species but it is not clear which version they used. I would also suggest for confirming the information of these genes using Solanaceae Genomics Network (www.solgenomics.net), which is solely dedicated to family Solanaceae. The discussion part of the manuscript could have been better as it focuses mainly on gene information of DFR not the functional roles and evolution of DFR. For this, authors may take advantage of Wang et al., (2022) The Plant Journal: 111,1096–1109.

Experimental design

Most missing part of study is the validation of DFR gene function to support the role of DFR in plants specifically in Solanaceae. Authors show some candidate gene function based on reported results of other species. These results could support to predict the gene function of candidate gene. I would highly suggest studying the function of identified genes using functional genomics tools such as transient/stable transformation of candidate genes. The VIGS or other similar tools are quite robust and can be used with ease for solanaceous plants.
• Abstract: A total of 42 putative DFR genes were systematically identified using bioinformatics to predict their protein properties, phylogenetic relationships, gene structure, conserved motifs, and cis-acting elements in the promoters: It is not clear how many genes are present in each species, please specify identified DFR genes in each species.
• Since gene expression varies greatly during plant development, please be specific about the age of the plant at the time the sample was taken for qRT-PCR. In a similar vein, what fruit growth stage was used for the expression analysis?
• I would suggest validating the qRT-PCR data with publicly available gene expression data of Solanaceae family (www.solgenomics.net).
• Figure 2 and 4: It is not clear genes mentioned in panel A belongs to which species? Please clarify.
• How were various genes classified using the gene nomenclature? It should be done in accordance with the Solanaceae family's recommended nomenclature.

Validity of the findings

I believe the validity of this research is ambiguous and imprecise without the crucial wet lab experimental data.

·

Basic reporting

Authors of this manuscript on 'Genome-wide identiûcation of putative dihydroûavonol 4-
reductase (DFR) gene family in eight Solanaceae species and expression analysis in Solanum lycopersicum' have presented results on the genome wide identification and characterization of DFR genes in Solanum lycopersicum. The work is presented very well. The authors identified 42 putative DFR genes by using bioinformatic tools and predicted their protein properties.
However, the following queries can be address for further improvement.
1. Represent the proper phylogenetic tree in Figure 1. Authors can use a circle representation of the phylogenetic tree instead of the table view.
2. Rewrite the figure legends for figure numbers 3, 5, and 6 with detailed data of figures.
3. Authors can add data on the cellular localization of the DFR genes.
4. “Discussion” should be written in a more logical and ordered way. It should be a reliable, deep, and more critical analysis of obtained results.
5. In Gene Expression Analysis of SlDFRs in Tomato, elaborate statistical analysis using students t-tests or ANOVA.
6. By analyzing promoter analysis data can perform qRT-PCR with some of the transcription factors identified in promoter analysis for understanding the better role of SIDFR genes.
7. Authors can also add information on relevance of differential tissue specific expression in different plant parts vis-a-vis the genes functions.

Experimental design

Good

Validity of the findings

Good but can be improved

·

Basic reporting

The manuscript submitted by Li et al. describes the dihydroûavonol 4-reductase (DFR) gene family in eight Solanaceae species for the first time. They have characterized the gene family members for their gene structure, protein features, cis regulator elements, evolutionary relationship and tissue specificity in Solanum lycopersicum by real time expression analysis.
The language of the manuscript is clear, however there are some grammatical and spelling errors that need to be taken care of. Some of them have been mentioned below, but whole manuscript needs to be rechecked again for language errors.
The background and introduction are well written with proper and sufficient references.
Figure legends need to be more elaborative. For example, in figure 1, if I am correct, the number in each column represent the number of genes in that species. This should be explained in figure legend for proper understanding. In figure 5, mention the full forms of Sl-R, Sl-S, etc in the figure legend.

Experimental design

There are hardly any experiments in the manuscript. Maximum is the bioinformatic analysis. But the bioinformatic analysis is carried out properly to characterize the gene family to maximum extent.

Validity of the findings

There is novelty in respect of the characterization of DFR gene family in tomato. However, I feel all the data is only based of bioinformatic analysis. For publication in PeerJ journal, I believe it is not enough. There should be more wet lab study to validate the bioinformatic data. At least the expression of DFR gene in different tissues could have been correlated with the anthocyanin content of the tissue to validate the expression of genes.

Additional comments

I found some minor errors, and few are mentioned here.
In the abstract, make all the scientific names in italicized font
Line 73: correct it as “asparagine”
Lines 145-146: You have written as “to measure the expression of tissue sample”. Here you are measuring the gene expression and not the expression of tissue sample. Please correct the sentence.
Line 301: correct it as “MEGA7”
Line 341, 344: correct the spelling of “melongena”.
Line 346: make the name of gene italicized VvDFR. Also look for gene names elsewhere in the manuscript if not italicized.

---

## Round 0.2 · accepted · Accept

Thank you for revising your manuscript.

Reviewer 1 ·

Basic reporting

Though, authors have addressed my comments partially but response to my major concern is still not satisfactory. This research without functional validation is not suitable for PeerJ publication. There are many approaches where VIGS or similar tools are used for functional validation of genes and studying anthocyanin biosynthesis. Few have been used in below papers;
The bZip transcription factor HY5 mediates CRY1a-induced anthocyanin biosynthesis in tomato. https://doi.org/10.1111/pce.13171
A Visual Reporter System for Virus-Induced Gene Silencing in Tomato Fruit Based on Anthocyanin Accumulation https://doi.org/10.1104/pp.109.139006

Experimental design

Partially improved.

Validity of the findings

No additional experiment is performed to validate the gene function.

·

Basic reporting

Revised version looks appropriate for acceptance

Experimental design

None

Validity of the findings

Very good; Revised version looks appropriate for acceptance

Additional comments

Revised version looks appropriate for acceptance

·

Basic reporting

Revised version is modified according to my comments and I have no further comments.

Experimental design

Revised version is modified according to my comments and I have no further comments.

Validity of the findings

Revised version is modified according to my comments and I have no further comments.

Additional comments

Revised version is modified according to my comments and I have no further comments.